# Bullous Pemphigoid: Trigger and Predisposing Factors

**DOI:** 10.3390/biom10101432

**Published:** 2020-10-10

**Authors:** Francesco Moro, Luca Fania, Jo Linda Maria Sinagra, Adele Salemme, Giovanni Di Zenzo

**Affiliations:** First Dermatology Clinic, IDI-IRCCS, Via Dei Monti di Creta 104, 00167 Rome, Italy; jlsinagra@hotmail.com (J.L.M.S.); a.salemme@idi.it (A.S.); g.dizenzo@idi.it (G.D.Z.)

**Keywords:** bullous pemphigoid, autoimmune bullous disease, trigger factors, predisposing factors, etiopathogenesis

## Abstract

Bullous pemphigoid (BP) is the most frequent autoimmune subepidermal blistering disease provoked by autoantibodies directed against two hemidesmosomal proteins: BP180 and BP230. Its pathogenesis depends on the interaction between predisposing factors, such as human leukocyte antigen (HLA) genes, comorbidities, aging, and trigger factors. Several trigger factors, such as drugs, thermal or electrical burns, surgical procedures, trauma, ultraviolet irradiation, radiotherapy, chemical preparations, transplants, and infections may induce or exacerbate BP disease. Identification of predisposing and trigger factors can increase the understanding of BP pathogenesis. Furthermore, an accurate anamnesis focused on the recognition of a possible trigger factor can improve prognosis by promptly removing it.

## 1. Introduction

Bullous pemphigoid (BP) is the most common autoimmune subepidermal blistering disease, affecting predominantly elderly people. It is characterized by generalized pruritic urticarial plaques and tense subepithelial blisters. BP autoantibodies are directed mainly against two hemidesmosomal proteins, BP180 (also termed type XVII collagen) and BP230, which are components of the dermo-epidermal junction (DEJ) [1]. In BP patients, BP180 and BP230 antigens are taken up by antigen-presenting cells, processed, bound to major histocompatibility complex class (MHC) II, and subsequently exposed on the cell surface. The recognition of these epitopes by T-cells induces the production of some cytokines and, subsequently, B cells are stimulated to produce autoantibodies (autoAbs). AutoAb isotypes against BP180 and BP230 are usually class G immunoglobulin (IgG), but IgE have been also implicated [2,3]. IgG autoAbs recognize mainly epitopes in the immunodominant region of BP180 termed NC16A [4]. IgG antibodies are directed also against the NH2- and COOH-termini of the cytoplasmic BP230 [5]. The binding of autoAbs leads to complement activation, recruitment of inflammatory cells, and release of proteolytic enzymes, activating an inflammatory cascade that could be even triggered by the activation of Th17 cells [6].

In the clinical suspect of BP, diagnosis bases on histopathologic analysis from the edge of a blister and direct immunofluorescence (DIF) on perilesional skin (Figure 1). Histology shows a DEJ detachment with a prevalent eosinophilic infiltration in the dermis, while DIF demonstrates a linear deposition of IgG and complement component 3 (C3) at the DEJ. Further, diagnostic approaches that could confirm and support the diagnosis of BP include indirect immunofluorescence on human skin (Figure 1), enzyme-linked immunosorbent assay (ELISA) based on BP180 and BP230 recombinant proteins and immunoblotting on keratinocyte extracts. First-line therapy of BP consists of topical or systemic corticosteroids. In case of refractory disease, or to minimize the adverse effects of chronic corticosteroid therapy, immunomodulatory drugs, such as azathioprine, mycophenolate mofetil, methotrexate, dapsone, or other drugs should be considered.

The etiopathogenesis of BP is largely unknown, but in several cases the occurrence or exacerbation of the disease has been reported in association with a specific “trigger factor”, such as drugs, physical factors, vaccines, infections, transplantations, and others. Prospective studies with detailed and precise anamnesis should be conducted in order to identify other hitherto unknown trigger factors. However, a specific trigger is “the straw that breaks the camel’s back”, acting together with several predisposing factors, such as HLA, comorbidities and aging that lead to the pathogenic cascade. The aim of the present review is to highlight the trigger and predisposing factors underlying the onset of BP.

## 2. Trigger Factors

### 2.1. Drugs

BP has been frequently associated with the assumption of systemic therapies. The putative drugs are antibiotics, beta-blockers, non-steroidal anti-inflammatory drugs (NSAIDs), diuretics and, more recently, anti-tumor necrosis factor (TNF)-α, dipeptidyl peptidase 4 inhibitors (DPP-4i), and immune checkpoint inhibitors targeting programmed cell death receptor 1 (PD-1) and its ligand (PD-L1) (Table 1).

The pathogenesis of drug induced BP (DIBP) is controversial and often difficult to understand and to demonstrate. As BP mostly affects elderly people, usually assuming several drugs, it is arduous to establish the triggering role of a specific medication. It has been hypothesized that the pathogenesis of DIBP is linked to the combination and interaction of various mechanisms, and not to a single one (Figure 2). According to the “two steps” theory, the interaction between two different drugs with similar molecular structures and the immune system could represent the first and second “hit” to trigger and to enhance an immune reaction [7]. Low molecular weight drugs could become immunogenic through non-covalent binding with molecules such as MHC and T-cell receptors (TCRs) inducing an immune response (Figure 2). Moreover, assuming the same drug during an initial immune response could stimulate its expansion. Another possible theory considers that drugs act as antigens, involving endogenous proteins in covalent binding. In this way, they could modify their antigenic properties exposing hidden antigenic sites or generating new antigens (Figure 2) [7]. A further possible trigger for the immune system is the interaction between the sulfhydryl groups in desmosomes and basement membrane zone (BMZ) and the sulfur-containing drugs that, by interrupting the DEJ integrity, could lead to the exposure of hidden antigenic sites. Another suggested mechanism in the pathogenesis of DIBP is the molecular mimicry. Several drugs bind micro RNA and other transcriptional and translational regulators analogously to a virus, and, therefore, it is possible that medications are mistaken with microbial antigens [7]. In a predisposed patient, the immune system could miss to identify a drug, favoring an activation of CD4^+^ T-cells and provoking the onset of an autoimmune disease (Figure 2) [7]. Some drugs can also cause the inactivation of endogenous regulatory processes involving T-cells. Medication like monoclonal antibodies against the immune checkpoint PD-1 and PD-L1 are actually an advanced therapy option against solid malignancies. Their mechanism consists in forcing immune checkpoint of T-cells. Therefore, the previously inhibited T-cells could amplify and improve the activity of the immune system resulting in the release of autoAbs, as anti-BP180 and anti-BP230, from previously depressed B-cells (Figure 2) [8].

As for diagnosis, we examined DIF results of 229 DIBP patients discussed in two retrospective case-control studies and in 42 case reports, and found 186 DIF positive patients, 15 DIF negative, and 28 with an unclear DIF result. Although in the literature some reviews and retrospective case-control studies do not always report DIF results, in the vast majority of DIBP patients, the diagnosis was obtained through the gold standard approach: DIF [9,10,11,12,13,14,15,16,17,18,19,20,21,22,23,24,25,26,27,28,29,30,31,32,33,34,35,36,37,38,39,40,41,42,43,44,45].

#### 2.1.1. Dipeptidyl Peptidase 4 Inhibitors (DPP-4i)

Evidence confirms that, considering all drug classes, previous use of DPP-4i carries the highest risk of developing a BP disease. DPP-4i, also termed gliptins, are a new class of oral hypoglycemic drugs, Food and Drug Administration approved in 2006 for type 2 diabetes. Sitagliptin was the first DPP-4i approved, but more than ten other DPP-4i are adopted in clinical practice. DPP-4i can be used in monotherapy or in association with other anti-hyperglycemic agents, such as metformin, sulfonylureas, thiazolidinediones, and basal insulin. The inhibition of DPP-4 improves the level of glucagon like peptide-1 (GLP-1) and glucose-dependent insulinotropic peptide (GIP) through the reduction of their catabolism induced by DPP-4. This process leads to increased secretion of insulin and lower secretion of glucagon that induce glycaemia reduction and glycemic control. Recently, it has been reported that DPP-4i are probably responsible for some cases of mucous membrane pemphigoids, a disease belonging to the pemphigoid group, and characterized by the prevalent involvement of mucous membranes [24]. Furthermore, a population-based study demonstrated an association of DPP-4i with multiple sclerosis, psoriasis, thyroiditis, BP, and inflammatory bowel disease [46].

The pathogenesis underlying the association between DPP-4i and BP is not clear. It is known that DPP-4 is a receptor of cell-surface plasminogen that activates plasminogen, leading to the formation of plasmin, which is a major serine protease that cleaves the NC16A domain of BP180. The inhibition of plasmin by DPP-4i could provoke alterations in the correct cleavage of BP180, modifying its antigenicity and function [47]. Furthermore, keratinocytes, endothelial cells, and T-cells express DPP-4, inhibition of which may increase the activity of pro-inflammatory cytokines, such as eotaxin, provoking a cutaneous eosinophil activation and blister formation [48].

Since 2011, the linkage between DPP-4-i and BP has been widely reported in literature including case reports, cases series, and other epidemiological studies. In that year, Pasmatzi and coworkers described two cases of BP induced by vildagliptin [25]. Moreover, one hundred cases of BP due to this DPP-4i, and to others, such as sitagliptin, linagliptin, saxagliptin, teneligliptin, anagliptin, and alogliptin, have been reported [25,26,27,28,29,30,31,32,34,35,36,37,38,49,50,51,52,53,54,55,56,57,58,59,60,61,62,63,64,65,66,67,68,69,70,71,72,73] (Table 1). Epidemiological evidence significantly supports the correlation of DPP-4i and BP onset. Early epidemiological studies based on data from European and French pharmacovigilance databases and showed the increased incidence of BP patients treated with DPP-4i [70,71]. Furthermore, in 2017, a Swiss case-control study investigated BP patients diagnosed between January 2007 and July 2013. Among 93 BP patients, 23 were diabetic. Nine of them were on treatment with DPP-4i (five with vildagliptin ± metformin, three with sitagliptin ± metformin, and one with vildagliptin/metformin). The authors concluded that DPP-4i use was more frequent in diabetic BP patients than controls (39.1 and 33.5%, respectively) but no statistical significance was found [73]. In 2017, Benzaquen and coworkers, in a case-control study on 61 patients affected by diabetes and BP, and 122 diabetic controls, during a period from 2014 to 2016, demonstrated that DPP-4i were associated with an increased risk(almost three-fold) for BP with highest adjusted odds ratio for vildagliptin [56]. In 2018, the first nationwide registry study reported that vildagliptin was associated with ten-fold elevated risk for BP, and that the mean latency from vildagliptin exposure to BP diagnosis was about 449 days [57]. Another case-control study conducted in Israeli on 82 BP patients confirmed that vildagliptin was the DPP-4i most commonly associated with BP onset, although even linagliptin had an increased risk for BP [59]. According to some authors that mucosal involvement is more common in DPP-4i associated BP than in classical BP [59].

Interestingly, some studies carried out in Japan by Izumi and coworkers, in 2016, suggested that cases of DPP-4i-associated BP presented a “non-inflammatory phenotype” characterized by small blisters, scant erythema, and a limited distribution of skin lesions, with spars eosinophilic infiltration in the dermis [47]. Furthermore, in these BP cases, autoAbs targeted mainly different immune-dominant BP180 epitopes (mid-portion of BP180, but not the NC16A domain) compared to classical BP. They reported that these “non- inflammatory” BP were related to a specific association with the human leukocyte antigen type (i.e., HLA-DQB1*03:01 allele) [74]. These findings explain how a predisposing factor, such as a specific HLA, could lead to a BP disease through a precipitating factor as the gliptin intake.

#### 2.1.2. TNF-α Inhibitors

Anti-tumor necrosis factor (TNF)-α monoclonal antibodies are a class of drugs widely used in the treatment of autoimmune and inflammatory dermatological diseases, such as psoriasis, hidradenitis suppurativa as an off-label therapy, in BP, and pemphigus vulgaris (PV). TNF-α has an important role in the pathogenesis of BP because it is secreted by mast cells during the inflammatory phase of the disease [1]. Furthermore, serum levels of TNF-α have been correlated with the severity and number of lesions in BP disease [75]. The greater use of this biologic drug in the last decades have also increased the number of cases reporting TNF-α inhibitors as the cause of BP [76]. Anti-TNF-α biologics that induced BP are etanercept [23,45], adalimumab [41,42,43,44], infliximab [40], and efalizumab [39]. BP healed in all the reported cases subsequently to the interruption of the biologic therapy. Regarding the pathogenesis of BP, it is necessary to understand if biologic therapy acts as a triggering factor or causes a hypersensitivity reaction or acts as a second hit on a previously deregulated immune system. It has been reported that the ability of an anti-TNF-α agent to induce or to treat an autoimmune disease depends on the level of interferon(INF)-γ and interleukin (IL)-4 [77]. Liu and coworkers reported that TNF-α secreted by eosinophils is able to produce a great amount of T helper (Th)1 and Th2 chemokines. In particular, TNF-α and INF-γ may induce the secretion of chemokine (C-X-C motif) 9 (CXCL9), CXCL10, and Th1 chemokines, while a TNF-α and IL-2 may produce C-C motif chemokine 12 (CCL12), CCL22 and Th2 chemokines [78]. Therefore, anti-TNF-α agents may have the capacity to suppress Th1 response in Th1 driven diseases, such as psoriasis, and, otherwise, Th2 response in Th2 driven diseases, such as BP, depending on the levels of INF-γ [77].

#### 2.1.3. Antibiotics

Antibiotics are a trigger for several autoimmune diseases and one of the first classes of medications associated with BP. In the elderly, bacterial infections are common, as antibiotic treatment is a common finding during a pharmacological anamnesis of elderly patients. However, antibiotic-induced BP patients are usually younger than classic BP patients, showing an early onset of BP in these specific patients. Several case reports in the literature report a clear and direct association between antibiotic therapy and BP onset [77]. The mechanisms to explain the BP induced by antibiotics are not completely clear, and various theories are proposed. Some theories relate to the chemical structure of the drugs. Drugs containing a sulfhydryl group, such as penicillin, amoxicillin, penicillamine, and cephalosporins, could induce a dysregulation in the immune system with direct damage on T-cells suppressor functions, resulting in the release of autoantibodies. In addition, thiol drugs could provoke direct, non-immunogenic damage to the dermo-epidermal junction, exposing new antigens to the immune system [10]. Other drugs containing a phenol ring in their structure, such as cephalosporin, could act as a hapten, inducing the formation of antibodies directed against the DEJ [10]. Ma and coworkers., in a case report of DIBP by levofloxacin, suggest a type IV hypersensitivity reaction T-cells mediated to explain the association between the drugs and the BP onset [13]. Other classes of antibiotics associated with BP onset are nidazole [22], rifampin [33], and aminoquinoline [9]. Moreover, antimycotic as actinomycin, dactinomycin, and griseofulvin could provoke BP with an unclear mechanism [77,79]. However, only a few people who undergo antibiotic therapy develop BP, suggesting that more than one trigger or predisposition factors are needed.

#### 2.1.4. Diuretics

Diuretics are one of the most common agents reported as triggering factors for BP disease. Elderly patients generally assume diuretic drugs due to chronic cardiovascular diseases. This could lead to several treatment challenges among the same drug family when a side effect occurs. BP is reported as an adverse drug reaction of various classes of diuretics, such as loop diuretics (furosemide and bumetanide) [80,81], thiazides [11,21], and potassium-sparing diuretics (the epithelial sodium channel blockers triamterene and the aldosterone antagonist spironolactone) [12,82].

#### 2.1.5. PD-1 and PD-L1 Immune-Checkpoint Blockade

Monoclonal antibodies against the immune checkpoint PD-1 and its ligand PD-L1 are a valid novel therapy for several malignancies, such as advanced solid tumors or metastatic melanoma. Pembrolizumab, such as other monoclonal antibodies against PD-1 and PD-L1, enhances T-cells mediated immune response. This could result in a disproportional activity of the immune system that could lead to various autoimmune disorders. One of the most commonly reported adverse effect of this drugs is a mild dermatological toxicity, such as pruritus and cutaneous rush, and, in some cases, BP.

The causal relation between BP and these drugs remains unclear. Some authors supposed that BP onset could be due to the effect of anti-PD1 in breaking down the PD1/PDL1 pathway’s protection against T-cell-mediated autoimmunity, while other authors attributed the role of anti-PD1 in enhancing the B-cell activated antigen-specific antibody response [83,84]. Furthermore, both T-cells and B cells express PD1 and PD-L1; therefore, pathogenic B cells may be activated by anti-PD1 therapy in a T-cell independent fashion [85]. The activation of B cells may favor the production of autoAbs against BP antigens, leading to the subepidermal cleft of BMZ. Other studies are needed to clarify the exact pathogenetic mechanism of these drugs in BP disease.

In 2015, Carlos and coworkers [14] reported the first case of BP in a patient who underwent therapy with pembrolizumab for metastatic melanoma. Subsequently, another five cases of BP linked with anti-PD-1 therapy were reported by Jour and coworkers [86]. Two of these cases have been associated to pembrolizumab and three to nivolumab. In 2016, the first case of BP induced by Durvalumab was reported [15] in a 78-year-old female under treatment for metastatic melanoma. Furthermore, two other cases of BP provoked by pembrolizumab were reported [16,17]. In 2018, Siegel and coworkers retrospectively reviewed the medical records of their institute, and reported seven cases of BP during therapy with an anti PD-1/PD-L1 [18]. In 2019, Sadik and coworkers reported an anti-the shed ectodomain of BP180 (LAD-1) IgG-positive, anti-BP180 NC16A IgG-negative BP in a patient treated with nivolumab for metastatic melanoma [19]. Furthermore, in 2020, a retrospective case-control study that evaluated the association of 12 BP patients who developed bullous disease after anti-PD-1 therapy, with improved tumor response, was reported [8].

#### 2.1.6. Neuroleptics

Many studies showed a correlation between neuroleptics and BP [87,88]. Varpuluoma and coworkers, analyzing Finland’s Care Register for Health Care database, reported that certain drugs, including periciazine, melperone, haloperidol, biperiden, and risperidone were associated with an elevated risk for BP [89]. Considering that, as treated in the comorbidities section, BP is more frequent in patients affected by neurological diseases. It is not clear if neuroleptic drugs or neurological disease or both can provoke a BP disease; therefore, further studies are needed to better elucidate this possible correlation.

#### 2.1.7. Contact Pemphigoid

The application of some preparations on the skin or mucous membrane could provoke BP. The irritant mechanism of these drugs on the skin, such as benzyl benzoate or the allergic contact hypersensitivity caused by the 5-fluorouracil, could be a trigger factor for BP onset [20].

### 2.2. Vaccines

BP has been associated with the administration of various vaccines, such as influenza, swine flu, tetanus toxoid, and herpes zoster virus, although some authors do not support this association [90,91]. The administration of vaccinations might activate the immune system, but the pathogenesis through which vaccines may provoke BP remain unknown. A relationship between vaccination and BP onset is complicated to demonstrate, considering the large use of vaccines and the frequent absence of BP relapse after additional vaccination. It could be hypothesized that, in predisposing subjects, the inflammation provoked by the vaccination may favor a disruption of the BMZ and unveil hidden epitopes/antigens that lead to a production of anti-basement membrane specific antibodies [77].

BP in a 90-year-old patient (twelve hours after the administration of an influenza vaccine) and BP disease in four elderly patients (one month after vaccination with a similar vaccine type) have been reported [92,93]. BP has arisen after vaccinations administered even in childhood and infants [94,95]. Guerra and coworkers recently reported a five-month-old female patient affected by BP after a first dose of hexavalent (diphtheria, tetanus, pertussis, poliomyelitis, hepatitis B, and Haemophilus influenzae B) and pneumococcus vaccination four weeks before the onset of skin lesions [94]. Although the majority of reported cases had a positive DIF, a close relationship between vaccination and BP onset is difficult to prove. In fact, epidemiological studies demonstrating this association are lacking, and the widespread use of vaccines, not only in infancy, suggest a possible random association.

### 2.3. Infections

There are different ways for pathogen agents to induce an autoimmune disease. A virus could affect B and T-cells, stimulating them respectively to autoAbs production, and Th1 to Th2 switch (this as a consequence of IL-4 and IL-10 elevate production). Furthermore, a pathogen can induce cross-reactive autoAbs sharing epitopes with host cells. For a virus, another way to induce autoimmunity is by directly infecting keratinocytes, inducing the expression of hidden epitopes or modifying those existing. Viruses could also insert in the envelope fragments of cells, producing new antigens [96,97]. Some authors have linked human herpes viruses (HHV), such as cytomegalovirus, Epstein-Barr virus, HHV-6, and HHV-8 with BP [97], finding virus DNA in blister fluid, and specific antibodies in the serum of patients [97,98]. Moreover, patients with BP are at an increased risk of developing viral infections, such as Herpes Zoster (HZ) [99]. Blaszek and coworkers found a higher prevalence of Torque Teno virus (TTV) in BP patients than in the control group, or in the pemphigus patients group [100]. This could suggest a role of TTV in the pathogenesis of BP. Human Immunodeficiency Virus (HIV) has been associated to BP in three case reports [101]. In all of them HIV infection preceded the BP onset [102,103,104]. Other agents associated with BP are Hepatitis B and C viruses [105], *Toxoplasma gondii* and *Helicobacter pylori* [96,98]. Thus, bacterial infections could also trigger BP disease, as reported in a 63-year-old man who developed localized BP on the left calf after two episodes of erysipelas [106]. Furthermore, parasitic diseases, such as Sarcoptes scabiei infestation, may trigger BP as a Koebner phenomenon, but it is very important to differentiate it from a bullous subtype of scabies that could mimic BP [107]. With few exceptions, where DIF was not performed for different reasons [101,102,103], in all the reported cases the diagnosis was confirmed by DIF.

### 2.4. Physical Factors

Triggering or exacerbation of BP has been reported after exposure to a variety of physical factors. Involved agents include injuries such as trauma, surgical interventions, thermal or electric burns, ultraviolet (UV) exposure, radiotherapy (RT) and, more recently, photodynamic therapy. Despite the frequent clinical and empirical observation of this relationship, reports are often anecdotal and large epidemiological data are lacking. Mai and coworkers, in a recent review, identified 147 published cases of BP triggered by external factors. Most commonly reported triggering factors in BP are: RT (25.4%), UV (25.1%), surgical procedures (37%), thermal or chemical burns (9.5%). Other factors (scratching, mechanical trauma, insect bite, dye injection) have been occasionally reported [31].

The association between BP and radiotherapy has been widely described [68,108,109,110]. Hung and coworkers recently published a case-control study on female patients with autoimmune bullous diseases (BP and PV). They found that a medical history of RT or breast cancer was associated with a higher risk to develop BP or PV [111]. A review by Nguyen and coworkers reported 29 published cases of BP provoked by radiotherapy; 86% of them were women, among whom 84% had breast cancer; 72% developed BP after RT (mean time of onset 15.8 months, median 5 months, range 2 weeks–5 months), while in 28%, BP arose during the RT cycle. The mean dose of inducing RT was 27.7 GY (range 20–46). At onset, the disease involved area of irradiation in 93% of patients and, subsequently, became generalized in 41% [110]. RT has been reported to trigger BP in association with anti-PD1 antibody nivolumab [112,113,114].

Several cases of BP triggered by thermal or chemical burns have been described [31,115,116,117,118]. In these cases, disease usually remains localized, has a favorable clinical course and rarely relapses. Mai and coworkers recently reported a case of BP triggered by thermal burn in a patient treated with a DPP4i, which became generalized and relapsed after a few months, despite drug suspension [31].

BP can arise on sites of surgical procedures, such as surgical wounds [31,109,119,120], skin graft [31,121,122,123], and ostomy [31,109,124,125]. The onset of BP on old scars has been reported [126].

BP can be induced by UV or phototherapy (mainly psoralen and ultraviolet A or ultraviolet B therapy) for other dermatologic diseases, usually psoriasis or Micosis fungoides [31,109,127,128,129,130]. Psoriatic patients are at higher risk for BP [131]. Even if the exact etiopathogenesis underlying this association is still not clear, it has been hypothesized that a role of abnormal psoriatic BMZ triggers BP [131,132,133]. As BP lesions usually appear on preexisting psoriatic lesions, it has been postulated that this mechanism could be triggered by antigenic changes induced by UVB and PUVA on psoriatic skin [134,135].

Photodynamic therapy, a technique utilized mainly for the treatment of no-melanoma skin cancers, has been indicated as a possible trigger factor in a small number of cases [136,137].

Overall data suggest that patients with BP induced by physical factors have a lower mean age and are predominantly females [109]. Disease can become generalized or remain localized to the site of injury. Despite the fact that the majority of cases of the disease arises after a time interval, varying from 1 month to 1 year from first exposure to triggering agent, onset can occur after several years [109].

In described cases, BP usually appears for the first time after exposure to the incriminated factor. The exacerbation of a previously diagnosed BP has been reported [138,139], while recurrence has rarely been described [31,140]. In the majority of BP patients (60–86%) associated to physical factors, the clinical diagnosis was confirmed by a positive DIF [109].

The pathogenetic mechanisms through which external agents could provoke or exacerbate BP are not clear. According to some authors, trauma induced tissue disruption leads to consequent exposure of otherwise masked antigens and development of BMZ-directed autoAbs [109]. Other authors suggest the pre-existence of low titers of circulating autoAbs in predisposed individuals. In these subjects, damaged tissue could release a variety of pro-inflammatory factors that may contribute to the recruitment of inflammatory cells and circulating antibodies, activation of granulocytes, and complement, leading to the formation of bullae [109]. This hypothesis seems to be confirmed by animal models, where UV irradiation leads to development of severe inflammation and bullae in subjects with circulating BP180 and/or BP230 autoAbs, but not in controls [141,142].

### 2.5. Transplantation

BP has been rarely reported in transplant patients. All reported cases were DIF positive and partially resistant to the systemic steroid therapy. In two cases, cutaneous lesions resolved after removal of the graft organ [143]. It could be hypothesized that autoAbs production in renal allografts could result from a chronic allogeneic process due to a cross-reactivity between the skin and the kidney. Otherwise, it should be considered that, in transplant patients, BP could be induced by drugs, such as immunosuppressants, which are necessary to avoid the rejection response. In the pathogenesis of BP, tacrolimus and other immunosuppressants that may reduce the expression of IL-2 gene could be involved in suppressing the number of Tregs and modifying the autoAb induction [143]. Furthermore, Kerkar and coworkers reported a case of BP after a liver transplantation, due to liver failure in a child with Coombs positive autoimmune hemolytic anemia and giant cell hepatitis [144].

### 2.6. Nutrition

No dietary triggers have been suspected of being involved in the induction of the BP [96]. One case of DIF positive dyshidrosiform pemphigoid induced by nickel in the diet has been reported, completely resolved after a nickel-free diet [145]. The 23-year-old woman had recurring itchy vesicular eruption of the hands for 5 years. The lesions completely resolved after prednisolone therapy combined with a low-nickel diet. After oral nickel challenge, mild itching and a few bullae occurred within 24 h. Pomponi and coworkers reported that BP patients showed a peculiar profile of IgE recognition toward some groups of allergens [146]. Specifically, a significant, higher prevalence of hen’s egg recognition was observed in patients with BP who had specific IgE to BP180 immunodominant region [146]. To assess the possible cross-reactivity between anti-NC16A IgE and hen’s egg, further studies are needed. Therefore, although few case reports suggest that some dietary factors can affect bullous skin diseases, additional studies of dietary manipulations, and of the effects of diet components on bullous dermatoses, are needed.

## 3. Predisposing Factors

### 3.1. Genetic Susceptibility

Analogously to other autoimmune diseases, polymorphism of some genes could have a role in BP onset. Considering that genetic susceptibility is inherited, it is very important to consider even the familiar genetic background. MHC genes, a highly polymorphic region located on the short arm of chromosome 6 (6p21), are an important component of the immune response. The three classes of molecules within the MHC are class I (i.e., HLA-A, -B, and -C), class II (i.e., HLA-DR, -DP, and -DQ), and class III (complement and cytokine genes) [97]. It has been reported, in population studies, that HLA class II alleles are associated with BP in several ethnic groups, including British, German, Japanese, Chinese, and Iranian populations [147,148,149,150]. Specifically, HLA-DQB1*03:01, belonging to the HLA class II allele, is associated with BP [147,150,151,152] and with distinct clinical variants [153,154,155]. Other studies suggest that HLA class I genes may be strictly associated with autoimmune diseases for its role in processing and presenting peptides to T-cells. An association with HLA class I genes has been found in autoimmune diseases such as pemphigus, multiple sclerosis, and type I diabetes, but not in BP [156,157,158,159,160]. Recently, Fang and coworkers studied HLA class I and HLA class II alleles with susceptibility to BP in the northern Chinese Han population. They reported that genetic susceptibility differences in ethnic groups are maintained in patients living away from their countries of ethnic origin, underlining the importance of genetic risk factors in BP [161]. In addition, in 2015 it was reported that polymorphisms in the mitochondrially encoded ATP synthase 8 gene are associated with susceptibility to BP in the German population [162].

### 3.2. Comorbidities

Association studies between BP and other diseases have been largely conducted (Table 2). A concomitant disease preceding BP occurrence could have a causal relationship with BP or with a predisposing factor linked with both BP and disease, such as HLA, aging, or immune system dysregulation.

#### 3.2.1. Neurologic Diseases

The association between BP and neurologic disease has been largely investigated. In 2010, two different case-control studies [163,164] found an overall prevalence of neurological disease respectively of 42.7–46% in BP patients vs. 19.1–11% of controls. Teixeira and coworkers found at least one neurologic diagnosis in 55.8% of BP patients vs. 20.5% of controls, stroke in 35.1% vs. 6.8%, dementia in 37.7% vs. 11.9%, and Parkinson’s disease in 5.1 vs. 1.1%, respectively. Several studies, conducted on larger scales, confirmed these findings and associated BP to different neurological conditions such as Parkinson’s disease, Alzheimer’s disease, multiple sclerosis, and stroke [165,166]. According to Chen and coworkers the association of BP and neurological diseases seems to be particularly strong with stroke (OR 3.30, 95% CI 3.03–3.60), dementia (OR 4.81, 95% CI 4.26–5.42), Parkinson’s disease (OR 3.49, 95% CI 3.03–3.60), epilepsy (OR 3.97, 95% CI 3.28–4.81), and schizophrenia (OR 2.56, 95% CI 1.52–4.30) [167]. Conversely, Langan and coworkers, in a large population-based case control study (868 cases vs. 3.453 controls) found that patients with dementia and Parkinson’s disease had a 3-fold higher probability of developing BP, while stroke and epilepsy were associated with a 2-fold increased odds of BP [165,168]. According to the majority of studies, the association with multiple sclerosis is less clear, even if, more recently, Kibsgaard and coworkers, in a large population-based cohort study, confirmed an overall higher frequency of neurological diseases, including multiple sclerosis, among patients with BP. Interestingly, the authors found that, during follow up, patients with BP had a higher hazard ratio of multiple sclerosis than controls (HR 9.4, 95% CI 4.9–18) [169].

Several hypotheses have been postulated to explain the association between BP and neurological disease. The existence of different isoforms of BPAG1 and BP230 has been demonstrated [170]. One of them, BPAG1-a, is expressed in neural tissue [170,171]. Circulating anti-BP180 can be found in patients with neurologic disease also in the absence of clinical signs of BP [172], and their levels have recently found to positively correlate with cognitive impairment in patients with BP and Alzheimer’s disease [173]. On the other side, Taghipour and coworkers did not find a significant difference in antibody profiles (anti-BP180 a/o anti-BP230), their titers between BP patients affected, or by a neurological disease [174]. More recently, Gornowicz-Porowska and coworkers found insignificant differences in autoAbs profiles between ethnic Pole BP patients with and without neurological disease [175]. Recently, BPAG2 has been demonstrated to be expressed in human brain [170,176], suggesting that this antigen could also play a pathogenetic role. One possible explanation is that brain damage subsequent to disease leads to the exposure of neural antigen and to the production of autoAbs that cross react with basal membrane antigens [170]. This is consistent with several studies, reporting that in 50.3–72% of patients, the onset of neurological disease precedes BP [164,167,177]. However, this is not a constant finding, as other authors reported a higher risk of developing a neurologic disease during the follow up of BP patients. According to Kibsgaard and coworkers, BP patients had a higher hazard ratio of multiple sclerosis during follow up than controls (HR 9.4, 95% CI 4.9–18) [169], while Brick and coworkers found a 8.56 HR for Parkinson disease and 2.02 for neurologic disorders in general [178]. No clear data on the prognostic factors of the association of BP with neurological disease are available, but some studies seem to indicate a higher mortality rate for this subgroup of patients [179]. Concluding, while the association between BP and neurodegenerative diseases seems accepted, the exact pathogenetic mechanisms are still not been fully explained.

#### 3.2.2. Other Autoimmune and Dermatologic Diseases

BP has been associated with various autoimmune diseases, but no large case-controlled studies have been performed regarding this possible relationship [180,181,182]. Taylor and coworkers studied the increased incidence of autoimmune disorders in BP patients, analyzing specific haplotype, such as HLA typing at the A, B, C, and DR loci, in 55 of the 108 BP patients. They concluded affirming that there is no increase incidence of autoimmune disorders in BP patients, and no specific haplotype is associated with a predisposing to this condition [182]. Lillicrap reported 15 cases of BP associated to rheumatoid arthritis [180,181]. Many authors suggested that this relationship is more than coincidental and could share a pathogenetic mechanism [183,184]. Furthermore, Savin confirmed this relationship reviewing 94 cases of BP, 11 of which were affected by rheumatoid arthritis [185]. BP has also been associated with lupus erythematosus, lichen planus, membranous nephropathy, pernicious anemia, thyroiditis, primary biliary cirrhosis, multiple sclerosis, psoriasis, and polymyositis [181,186]. Furthermore, some autoimmune phenomena, such as a high levels of antinuclear factor, anti-thyroid antibodies, anti-smooth muscle antibodies, and rheumatoid factor, have been described in cases of BP and cicatricial pemphigoid [187].

Psoriasis and BP are frequently associated, despite this association has only recently been studied on large populations [188]. In 1985, Grattan and coworkers, in a small case-control study, found a higher prevalence of psoriasis among BP patients than in controls. More recently, in a case control study of 51,800 psoriasis patients, Tsai and coworkers found that the prevalence of BP was higher among psoriasis patients than controls (OR 14.8) [189]. This finding have been further confirmed by other studies [167,190]. According to Ohata and coworkers, association between psoriasis and BP seems more frequent among males, and in most patient psoriasis preceded BP onset. Among bullous diseases, BP was the most prevalent (63.4%) followed by anti-laminin γ1 pemphigoid (37.2%) [191]. Ho and coworkers found that patients with psoriasis have a 3.05 fold higher risk of BP [131]. Several hypothesis explaining this association have been advanced. BP and psoriasis do not share any common HLA or otherwise genetic susceptibility. Considering that psoriasis precedes BP in the large majority of patients, and, therefore, the latter often arises on preceding psoriatic lesions, it has been postulated that the onset of BP could be a consequence of epigenetic events related to the psoriatic inflammatory cascade [188]. Changes at the BMZ in psoriasis may be responsible for the heterogeneous antibody response, and may trigger bullous disease [132,133]. Increased serum CCL28 levels and the possible role of Th17 cells in patients with psoriasis and BP may be another link for their association. There is evidence that Th17 cells and IL-17 may be involved in the production of pro-inflammatory cytokines, matrix metalloproteinases, neutrophils, and eosinophils, all of which are important pathogenic factors in BP. Furthermore, the presence of Th17 lymphocytes in conjunctival biopsies was significantly increased [132,188].

#### 3.2.3. Neoplasms

The association between BP and malignancies is controversial [192,193]. Older case control studies failed to demonstrate a higher incidence of malignant tumors among patients affected by BP. Recently, Lucariello and coworkers [194], in a meta-analysis comparing the results from 16 studies, found an 11% rate of malignancies among patients with BP. The authors found a statistically significant association between advanced age and the rate of malignancy. Several studies suggest an association with hematologic malignancies [195]. Conversely, in a large retrospective study, Ong and coworkers did not find an overall association between cancer and BP, except for some specific tumors (kidney cancer, laryngeal cancer, lymphoid leukemia) [196]. Several hypotheses have been proposed to explain this association. According to the more reliable, some autoantibodies directed towards tumor antigens could cross-react with basal membrane antigens, eliciting blister formation. One possible antigen could be laminin-332, a protein involved in attachment between keratinocytes and the basal cell membrane, expressed also by other solid malignancies (breast, pancreas, colon, and lung cancer). Other hypotheses include the production of hormone-like substances that disrupt the basal membrane provoking the development of anti-basement membrane antibodies, the tumorigenic activity of external factors that could contextually damage BMZ, and the tumor production of cross-reactive antigen towards BMZ. In conclusion, the association between BP and tumors is far from been definitively confirmed. In this view, some authors suggest performing oncological screening only in some restricted cases.

#### 3.2.4. Cardiovascular Diseases

Several studies seem to indicate an association between BP and cardiovascular diseases [197,198]. In a retrospective cohort study, Roujeau and coworkers demonstrated that the number of deaths from cardiovascular diseases in patients diagnosed with BP was higher than expected in the general population of the same age and sex [199]. In a case series, Echigo and coworkers reported that thromboembolism occurred in 7 of 20 (35%) patients with autoimmune blistering disease, including BP, involving various organs including brain and lung [200]. A population-based study found that BP patients are three times more likely to develop pulmonary embolism [201]. Ya-Wen and coworkers found that BP patients had a 2-fold higher incidence of stroke than controls (22.8 vs. 11.4%) [202]. Several possible reasons have been proposed to explain the association between BP and cardiovascular diseases. One possible explanation of this link is the hypercoagulable state of BP patients, due to inhibition of fibrinolysis and activation of coagulation [203]. The hypercoagulation state can be related to antiphospholipid antibodies (aPLs). aPLs are a heterogeneous group of antibodies, including lupus anticoagulant, anti-cardiolipin antibody, and anti-2 glycoprotein I antibody [204]. These antibodies frequently appear in systemic lupus erythematosus [205], but have also been detected in organ-specific autoimmune diseases, such as insulin-dependent diabetes mellitus [206,207], myasthenia gravis [208], autoimmune thyroid diseases [207,209], and inflammatory bowel disease [210,211]. A study shows that aPLs were detected in 10 of 20 patients with autoimmune blistering disease, including 2 of 4 patients with BP [200]. aPLs can produce a prothrombotic state through inhibition of natural anticoagulant pathways, inhibition of fibrinolytic mechanisms, and induction of endothelial cell pro-coagulant activity. These processes predispose patients to thrombotic events, particularly ischemic stroke [212,213]. A recent multicenter cohort study on 432 patients showed that the risk of venous thromboembolism (VTE) is 4-fold higher in BP patients when compared to the general population of the same age and sex [214]. More specifically, VTE risk increases up to 15 times during the acute phase of the disease, indicating a close link between the inflammatory state occurring in active BP and thrombosis. Interestingly, the VTE risk dropped 1.5 times during clinical remission, indicating that a good control of the disease dramatically reduces the occurrence of thrombotic events. It has been consistently reported that the risk of thrombosis is increased in patients with BP [202,215,216]. Cugno and coworkers have focused on the relationships among immune response, inflammation, and blood coagulation in bullous pemphigoid [217]. As is known, the dermo-epidermal detachment is due to the interaction of autoAbs with BP180 and BP230 hemidesmosomal antigens, followed by complement activation and leukocyte infiltration. Autoreactive T lymphocytes cooperate with B lymphocytes in the autoAb production; in addition, they release cytokines, most notably IL-5 and IL-16, and other soluble factors responsible for the recruitment and activation of eosinophils. Indeed elevated concentrations of secretory granules, such as eosinophil cationic protein (ECP) [218,219] as well as increased levels of IL-5 [220,221] and IL-16 [222], had been found in blister fluid and sera of patients with BP. In lesional skin, eosinophils produce and release metalloproteinases, elastase, and gelatinase, which contribute to tissue damage, degrading BP180 and cleaving the dermal-epidermal junction [223,224,225]. Moreover, eosinophils strongly express tissue factor, which is the main initiator of the coagulation cascade (factors VII, X, VII, V, and prothrombin) leading to generation of thrombin [226,227]. This increases the permeability of blood vessels [228], thus favoring the transendothelial migration of inflammatory cells and their accumulation in the skin [217]. Neutrophils contribute to the pathogenesis of BP by releasing reactive oxygen species (ROS) and proteases. Activated proteases, in turn, act on protease-activated receptors (PARs) and induce the expression of various pro-inflammatory cytokines; this cross-talk between inflammation and coagulation amplifies and maintains the activation of both systems [227]. It is noteworthy that BP patients have high levels of coagulation activation markers, such as prothrombin fragment F1+2, indicating thrombin generation, and D-dimer, indicating fibrin degradation, in plasma samples other than in blister fluid [226,227,229,230]. A study on a group of patients with active BP showed that fibrinolysis is inhibited, mainly due to an increase in the plasma levels of plasminogen activator inhibitor type 1 (PAI-1) activity and antigen [203]. The high thrombotic tendency found in patients with BP, especially during the acute phase of the disease, implies some practical considerations: whether adding VTE prophylaxis to the immunosuppressive treatments could affect the thrombotic risk of BP patients, and if clinical trials on efficacy and safety of antithrombotic drugs administered in the acute phase of the disease could provide some insight into their clinical relevance.

### 3.3. Aging

Age-dependent changes affect both the adaptative and innate immune response. The immunosenescence is the dysregulation process of the immune system resulting in three main events: (1) reduced immune response; (2) increased autoAbs production; (3) inflammaging (chronic, sterile, low grade inflammation background) (Figure 3) [231]. With aging, B cell production and development are reduced due to several factors such as modified cellularity of the bone marrow, epigenetics, and telomeres modification of hematopoietic stem cells (HSCs), an impaired transition from pro B cell to pre B cell, decreased number of naive B cell, fall in production, a release of IL-7 and recombination of heavy chain gene [232,233].

Response to new antigens is reduced, but the serum level of IgG1, IgG2, and IgG4 is raised (Figure 3). In the elderly, we can observe an increase in monoclonal low-affinity immunoglobulins produced by CD19^+^ CD5^+^ cells without the involvement of T-cells [234].

An impairment in immunoglobulins class switching results from a dysregulation in the expression and activation of phosphotyrosine kinases and protein kinase C [235].

Memory B cell number rises, but with a reduced affinity of antibodies and decreased diversity.

In the elderly a new population of B cells, called age-related B cells (ABCs), develops. ABCs do not express CD21, but they express CD11c and CD11b. ABCs proliferate after stimulation with toll like receptor 7 (TLR7) and TLR9, and do not proliferate after B cell receptor (BCR) stimulation [236,237].

A sustained population of ABCs appear in the elderly after simultaneous stimulation of TLRs, BCR, and inflammatory cytokines [238]. This phenomenon is observed during chronic infections such as HIV or tuberculosis, and chronic autoimmune diseases such as rheumatoid arthritis and systemic lupus erythematosus [239]. Their involvement in the development of autoimmune diseases is suggested by observing a reduced autoAb titer in animal models with ABCs depletion [240].

In parallel, T-cells dysregulation is a consequence of several events. First, the thymic involution and atrophy followed by genetic and epigenetic alterations of the hematopoietic stem cells. In addition to this, a chronic antigenic stimulation caused by persistent viral infections can wear out the immune system [231,241].

The first significant modification is a general depletion of T-cell CD3^+^, a depletion in naive CD45RA^+^, a decrease in the CD4/CD8 ratio, lower proliferative capacity, and T-cell repertoire diversity. Although Th2 activity remains unchanged, Th1 cell activity decreases [231].

With aging, we can observe a switch in surface receptors of terminal differentiated CD8^+^ T-cells towards natural killer cells, losing co-stimulatory receptors, and expressing NK-associated receptors (i.e., KIR, NKG2D) [242].

In tissues of elderly, a higher number of non-functional memory T-cells, such as CD4^+^ CD45RO^+^ and CD8^+^ CD45RO^+^ is usually observed [243].

Aging can cause a decrease in regulatory T-cells population, i.e., CD4^+^ CD25^+^ T-cells. This modification could induce tolerance towards the development of autoimmune processes (Figure 3) [244].

Dendritic cells number decrease; their capability to stimulate lymphocytes and antigen processing and presentation is compromised. Aging negatively affects the ability of dendritic cells of chemotaxis, endocytosis, and migrations. Moreover, the function of the TLR is compromised [241,243]. Macrophage activations and functions decrease due to a dysregulation in the TLR1 and TLR4 signaling cascade. Decreased phagocytosis, chemotaxis, and lower expression of MHC-II can be observed. Moreover, the production and release of reactive oxygen species are compromised with aging [245].

Neutrophils are subject to changes during immunosenescence, not in absolute number, which remains unchanged, but in functionality, due to a dysregulation of signaling transduction cascades. A decreased ability of chemotaxis and phagocytosis, as well as a compromised production and release of superoxide anion radical, is observed [246].

Despite high autoAbs levels in the elderly, the incidence of autoimmune diseases in this population is not high, as expected, except for some diseases, such as pemphigus, bullous pemphigoid, polymyalgia rheumatica, and temporal arteritis [247]. The meaning of this relevance in a selective choice of diseases remains to be clarified.

## 4. Conclusions

Predisposing factors of BP development include genetics, comorbidities, and aging. Precipitating factors, such as drugs, vaccines, infections, physical factors and transplantation, refer to a specific event or trigger that could induce or exacerbate BP disease in the context of several predisposing factors. The awareness of the possible role of both groups of factors can improve the understanding of BP pathogenesis and lay the basis for future studies aimed at identifying the relationship between them.

## Figures and Tables

**Figure 1 biomolecules-10-01432-f001:**
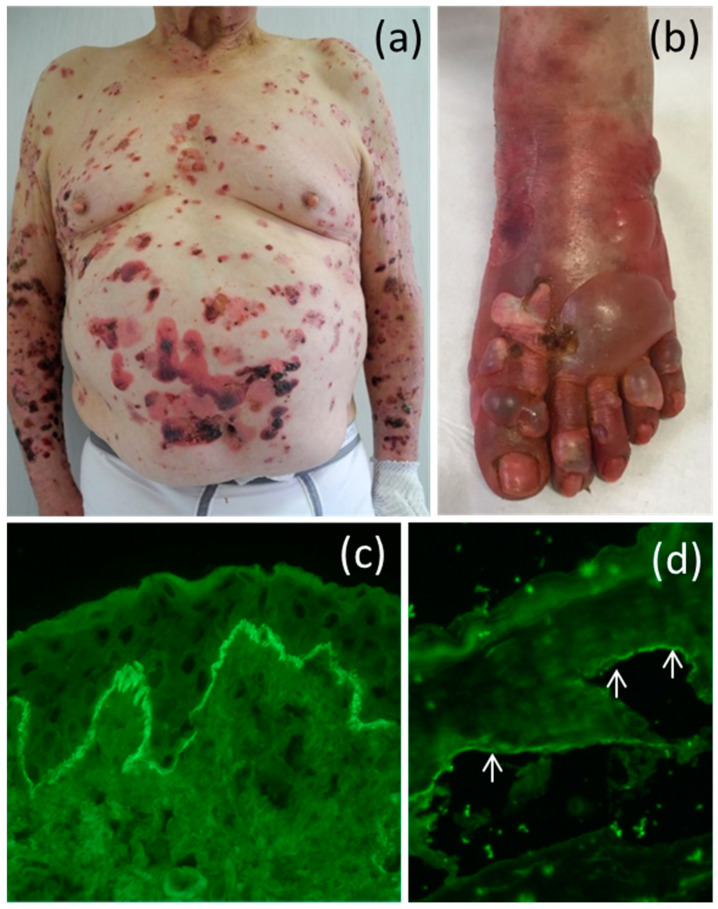
Clinical and immunological features of patients affected by gliptin-induced bullous pemphigoid (**a**,**b**) clinical presentation of two gliptin-associated bullous pemphigoid (BP) patients. Large tense blisters on erythematous plaques on trunk, arms, and foot are present. (**c**) Linear deposits of IgG at the dermal-epidermal junction by direct immunofluorescence microscopy. (**d**) IgG labeling of epidermal side (pointed by arrows) on human salt-split skin by indirect immunofluorescence microscopy.

**Figure 2 biomolecules-10-01432-f002:**
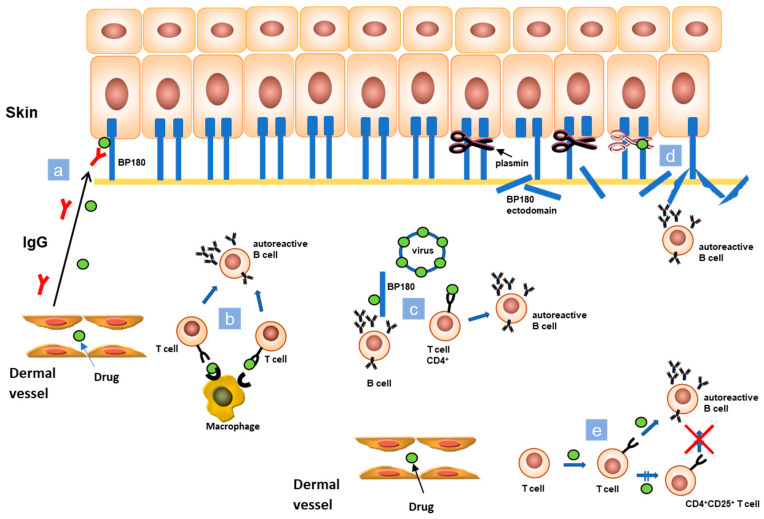
Schematic representation of proposed mechanisms of drug induced BP. (**a**) Drugs can act as antigens, involving endogenous proteins in covalent binding. They could modify their antigenic properties exposing hidden antigenic sites or generating new antigens. (**b**) Low molecular weight drugs could become immunogenic through non-covalent binding with molecules, such as major histocompatibility complex class (MHC) and T-cell receptors (TCRs) causing an immune response. (**c**) Another suggested mechanism at the basis of BP induction is molecular mimicry. It is possible that medications are mistaken with microbial antigens. This could also lead to activation of CD4^+^ T-cells and initiation of the autoimmune cascade. (**d**) The inhibition of plasmin by dipeptidyl peptidase 4 inhibitors (DPP-4i) could provoke alterations in the correct cleavage of BP180, modifying its antigenicity. (**e**) Some drugs can also cause the inactivation of endogenous regulatory processes involving T-cells or force immune checkpoint of T-cells, such as for antibodies against the immune checkpoint PD-1 and PD-L1.

**Figure 3 biomolecules-10-01432-f003:**
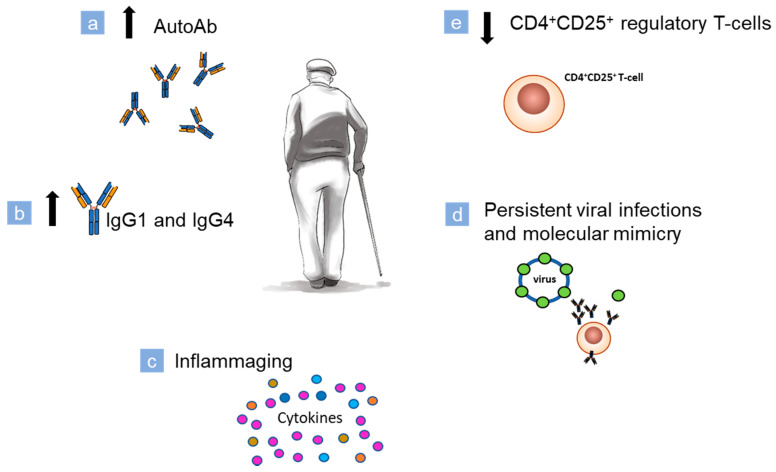
Aging-associated mechanisms that could contribute to the induction of BP onset. Ageing is accompanied by a process of remodeling/restructuring that involves immune system termed immunosenescence. (**a**) In the elderly, the production of low-affinity immunoglobulins increases, and (**b**) serum levels of IgG1 and IgG4 raise; (**c**) a chronic, sterile, low grade inflammation background is typical of the elderly. This condition (of basal cytokines production) could act as a stimulus for the autoimmunity onset; (**d**) in the elderly, infections are common and persistent. In a dysregulated immune system, peptides of an infective agent could lead to a cross-reaction against self-peptides with similar sequences. (**e**)The compartment of regulatory CD4^+^ CD25^+^ T-cells tends to decrease. These regulatory cells are able to counteract the occurrence of autoimmune events. Thus, a decline in their number could favor the development of autoimmune processes.

**Table 1 biomolecules-10-01432-t001:** Drugs associated with BP onset.

Immune Checkpoint Inhibitors Targeting Programmed Cell Death Protein 1 (PD-1) and Programmed Cell Death-Ligand 1 (PD-L1)	Antibiotics	Calcium Channel Blockers	Angiotensin-Converting-Enzyme Inhibitors (ACE Inhibitors)	β-Blockers	NSAID	Salicylates	Dipeptidyl Peptidase 4 Inhibitors	Diuretics	Others Antidiabetics	Anti TNF-α	Others
Pembrolizumab	Actinomycin	Amlodipine	CapTopril	Atenolol	Azapropazone	Aspirin	Sitagliptin	Furosemide	Tolbutamide	Adalimumab	Arsenic
Nivolumab	Amoxicillin	Nifedipine	Enalapril	Nadolol	Colecoxib	Sulphasalazine	Vildagliptin	Spironolactone		Efalizumab	Doxepin
Durvalumab	Ampicillin		Lisinopril	Practolol	Diclofenac (topical)	Salicylazosulphapyride	Alogliptin	Bumetanide		Etanercept	Clonidine
	Cephalexin			Angiotens. II antagonists	Ibuprofen		Linagliptin			Infliximab	Erlotinib
	Ciprofloxacin			Losartan	Mefenamic acid		TeneLIgliptin				Escitalopram
	Chloroquine				Phenacetin		Saxagliptin				Everolimus
	Dactinomycin						Anagliptin				Fluoxetine
	Griseofulvin										Flupenthixol
	Levofloxacin										Gabapentine
	Metronidazol										Galantamine hydrobromide
	Penicilline										Gold thiosulphate
	Rifampicin										Interleukin-2
											Iodinate contrast (IV iodine + etanercept)
											Levetiracetam
											Methyldopa
											Methotrexate
											Terbinafine
											Thiopronin
											Omeprazole
											Psoralens with UVA
											Placental extracts
											Potassium iodide
											Risperidone
											Rosuvastatin
											Sulphonamide
											Ustekinumab

**Table 2 biomolecules-10-01432-t002:** Bullous pemphigoid comorbidities.

Neurologic Diseases	Autoimmune Diseases	Neoplasms	Cardiovascular Diseases
Stroke	Psoriasis	Kidney cancer	Thromboembolism
Dementia	Rheumatoid arthritis	Laryngeal cancer	Stroke
Parkinson’s disease	Lupus erythematosus	Hematologic malignancies	Venous thromboembolism
Alzheimer’s disease	Lichen planus		Pulmonary embolism
Multiple sclerosis	Membranous nephropathy		
Epilepsy	Pernicious anemia		
Schizophrenia	Primary biliary cirrhosis		
	Thyroiditis		
	Multiple sclerosis		
	Polymyositis

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
