# Peer review of "Bullous Pemphigoid: Trigger and Predisposing Factors"

_biomolecules, 2020, doi:10.3390/biom10101432_

Round 1

Reviewer 1 Report

Moro et al. summarized a recent understanding of BP with trigger and predisposing factors.  The description of the manuscript is succinct but comprehensive. Although several mistakes in the English language were found, still, it is acceptable for me.

I strongly think that several tables and/or cartoons which shows possible speculative mechanisms of action for drugs to BP pathogenesis.  It can be done to attract readers, especially for Drug vs. BP and for the aging part.

BP180 should be BP180/type XVII collagen in the first part.

Errors;

P1L31; favours

P1L41;demonstrates

P5L19; double periods

P5L28; No periods

P6L93; No periods

P7L133, P9L214,216; Do not capitalize for Authors (several others were found throughout text)

P11 L316; no double spaces after "patients"

P14L464; Tool must be the mistake

P14L483; Site Ponnappan's paper correctly

P14 L499; Do not capitalize Pemphigus and Bullous Pemphigoid

Author Response

Moro et al. summarized a recent understanding of BP with trigger and predisposing factors.  The description of the manuscript is succinct but comprehensive. Although several mistakes in the English language were found, still, it is acceptable for me.

We thank the reviewer, we have carefully revised the manuscript and perform several changes to improve English language.

I strongly think that several tables and/or cartoons which shows possible speculative mechanisms of action for drugs to BP pathogenesis.  It can be done to attract readers, especially for Drug vs. BP and for the aging part.

We agree with the reviewer. We have prepared 2 more figures to summarize possible mechanisms of BP induction for drugs (Figure 2) and aging (Figure 3). Furthermore we have added a paragraph for antibiotics in the drugs section.

BP180 should be BP180/type XVII collagen in the first part.

We have accordingly modified the Introduction section.

Errors;

P1L31; favours

P1L41;demonstrates

P5L19; double periods

P5L28; No periods

P6L93; No periods

P7L133, P9L214,216; Do not capitalize for Authors (several others were found throughout text)

P11 L316; no double spaces after "patients"

P14L464; Tool must be the mistake

P14L483; Site Ponnappan's paper correctly

P14 L499; Do not capitalize Pemphigus and Bullous Pemphigoid

We have carefully revised the manuscript correcting all the errors indicated.

Reviewer 2 Report

The authors here present an excellent summary of BP triggers and comorbidities.

I recommend careful editing of the language (see for example Table 1, "Dypeptidil", "b" instead of greek letter β, page 5: "mucousal", etc.).

I would also be happy to see a (supplementary) table where all triggers are listed alongside published DIF microscopy results, e.g., is there any trigger where also DIF was positive, or, for which triggers is the DIF positive, and for which it is not?

Author Response

The authors here present an excellent summary of BP triggers and comorbidities.

We thank the reviewer for his/her positive comments on our manuscript.

I recommend careful editing of the language (see for example Table 1, "Dypeptidil", "b" instead of greek letter β, page 5: "mucousal", etc.).

We thank the reviewer for this suggestion. We have carefully revised the manuscript and perform several changes to improve English language.

I would also be happy to see a (supplementary) table where all triggers are listed alongside published DIF microscopy results, e.g., is there any trigger where also DIF was positive, or, for which triggers is the DIF positive, and for which it is not?

We completely agree with the reviewer. Information on DIF are crucial to evaluate the relation between triggers and BP. In fact, DIF is the diagnostic gold standard for BP and without positive DIF the relationship becomes not significant. Thus, we have summarized and discussed the collected information on DIF in the triggering factors section: pages3-4 lines119-123; page10 lines 185-188: page 11 lines 212-213; page 12 lines 257-258, 271, 286. However, we didn’t make a table that would be really huge with more than 500 cases to mention, also in the reference list. We hope that this choice may be accepted by reviewer.

Round 2

Reviewer 2 Report

Changes made by the authors are acceptable.

Author Response

We thank the reviewers for considering our changes acceptable.

Dear Editor,

thank you for giving us the opportunity to revise and improve our manuscript.  

Please find our point-by-point response to the reviewers’ comments/suggestions. 

All changes are shown as tracking changes in the revised manuscript.  

We have carefully followed all the suggestions/modifications of the reviewers.

We hope that our manuscript will now be suitable for publication. 

Best regards, 

Francesco Moro

Reviewer(s)'/Editor Comments to Author:

Major point: Basic immunology errors in the revised manuscript -see Fig. 3 (and legend thereof). 

In Figure 3:

In the figure per se: Change "Regulatory cells" to CD4+CD25+ regulatory T-cells because there are over 20 types of regulatory cells and the authors only mention CD4+CD25+.
In the legend of Fig.3: Change (5) to something that makes sense, regarding lower numbers (and/or function?) of regulatory T-cells and loss of tolerance.

We thank the reviewer for the suggestions. We changed "Regulatory cells" to CD4+CD25+ regulatory T-cells in the figure 3. In the legend of figure3 at point (5), now changed in (e), we modify the sentence in “The compartment of regulatory CD4+ CD25+ T-cells tends to decrease. These regulatory cells are able to counteract the occurrence of autoimmune events. Thus a decline in their number could favor the development of autoimmune processes”. We hope this new sentence could correctly clarify how the hypotized decrease of CD4+CD25+ regulatory T-cells number, occurrig with ageing, could promote autoimmunity. However, the mechanism of regulatory T-cells number decrease in the induction of autoimmunity is not yet clarified.

Minor point: Language

We thank the reviewer for the suggestion. We improved the english language in our manuscript

throughout the whole text.
